# Brain Perception of Different Oils on Appetite Regulation: An Anorectic Gene Expression Pattern in the Hypothalamus Dependent on the Vagus Nerve

**DOI:** 10.3390/nu16152397

**Published:** 2024-07-24

**Authors:** Gele de Carvalho Araújo Lopes, Brenda Caroline Rodrigues Miranda, João Orlando Piauilino Ferreira Lima, Jorddam Almondes Martins, Athanara Alves de Sousa, Taline Alves Nobre, Juliana Soares Severo, Tiago Eugênio Oliveira da Silva, Milessa da Silva Afonso, Joana Darc Carola Correia Lima, Emidio Marques de Matos Neto, Lucillia Rabelo de Oliveira Torres, Dennys Esper Cintra, Ana Maria Lottenberg, Marília Seelaender, Moisés Tolentino Bento da Silva, Francisco Leonardo Torres-Leal

**Affiliations:** 1Metabolic Diseases, Exercise and Nutrition Research Group (DOMEN), Laboratory of Metabolic Diseases Glauto Tuquarre, Department of Biophysics and Physiology, Center for Health Sciences, Federal University of Piaui, Teresina 64049-550, PI, Brazil; gelecarvalho@gmail.com (G.d.C.A.L.); bcrm21@gmail.com (B.C.R.M.); joaoopfl@gmail.com (J.O.P.F.L.); jorddamalmondes@ufpi.edu.br (J.A.M.); athanaraalves@ufpi.edu.br (A.A.d.S.); taline.nobre@ufpi.edu.br (T.A.N.); julianasevero@ufpi.edu.br (J.S.S.); 2Department of Physiology and Biophysics, Institute of Biomedical Sciences, University of São Paulo, São Paulo 17012-900, SP, Brazil; eugenio_tiago@usp.br; 3Department of Inflammation and Fibrosis, Gilead Sciences, Foster City, CA 94404, USA; milessafonso@gmail.com; 4Cancer Metabolism Research Group, Department of Surgery and LIM26-HCFMUSP, Faculty of Medicine, University of São Paulo, São Paulo 17012-900, SP, Brazil; joana.carola14@gmail.com (J.D.C.C.L.); seelaender@usp.br (M.S.); 5Department of Physical Education, Federal University of Piauí, Teresina 64049-550, PI, Brazil; emidiomatos@ufpi.edu.br; 6Federal Institute of Education, Science and Technology of Maranhão, Caxias 65030-005, MA, Brazil; lucillia.rabelo@gmail.com; 7Laboratory of Nutritional Genomics, University of Campinas, Campinas 13083-855, SP, Brazil; dcintra@yahoo.com; 8Nutrigenomics and Lipids Research Center, CELN, School of Applied Sciences University of Campinas, São Paulo 13083-970, SP, Brazil; analottenberg@gmail.com; 9Hospital Israelita Albert Einstein (HIAE), São Paulo 05652-900, SP, Brazil; 10Laboratório de Lípides (LIM10), Hospital das Clínicas HCFMUSP, Faculdade de Medicina, University of São Paulo, São Paulo 17012-900, SP, Brazil; 11Institute of Biomedical Sciences Abel Salazar, Center for Drug Discovery and Innovative Medicines, Laboratory of Physiology, Department of Immuno-Physiology and Pharmacology, University of Porto, 4099-002 Porto, Portugal

**Keywords:** fatty acids, appetite regulation, gastrointestinal motility, vagus nerve

## Abstract

(1) Background: We examined the effect of the acute administration of olive oil (EVOO), linseed oil (GLO), soybean oil (SO), and palm oil (PO) on gastric motility and appetite in rats. (2) Methods: We assessed food intake, gastric retention (GR), and gene expression in all groups. (3) Results: Both EVOO and GLO were found to enhance the rate of stomach retention, leading to a decrease in hunger. On the other hand, the reduction in food intake caused by SO was accompanied by delayed effects on stomach retention. PO caused an alteration in the mRNA expression of NPY, POMC, and CART. Although PO increased stomach retention after 180 min, it did not affect food intake. It was subsequently verified that the absence of an autonomic reaction did not nullify the influence of EVOO in reducing food consumption. Moreover, in the absence of parasympathetic responses, animals that received PO exhibited a significant decrease in food consumption, probably mediated by lower NPY expression. (4) Conclusions: This study discovered that different oils induce various effects on parameters related to food consumption. Specifically, EVOO reduces food consumption primarily through its impact on the gastrointestinal tract, making it a recommended adjunct for weight loss. Conversely, the intake of PO limits food consumption in the absence of an autonomic reaction, but it is not advised due to its contribution to the development of cardiometabolic disorders.

## 1. Introduction

Different dietary lipids may have a major influence on the complicated physiological process of food intake regulation [1,2,3]. Understanding the impact of these different lipids on the processes that control food intake is critical for unraveling appetite control pathways and better understanding their relevance to metabolic health [4,5,6]. Despite extensive research on the effect of dietary lipids on satiety and appetite regulation, little is known about the acute effects of different types of oil excess in the stomach and their effect on food intake regulation.

The theoretical and practical consequences of the predicted outcomes of this investigation are critical. Examining the acute effects of lipid ingestion on the stomach can provide a deeper understanding of the physiological mechanisms involved in food intake regulation, which could significantly impact the development of therapeutic strategies and interventions for managing body weight and treating metabolic disorders [7,8]. Furthermore, investigating the short-term consequences of too much oil in the stomach may give vital information about how structural changes in the gastrointestinal system affect short-term hunger [9,10]. This physiological approach offers unique and complementary perspectives on molecular research.

Previous research has mostly concentrated on the long-term effects of dietary lipids on hunger control, with little attention paid to transitory effects, notably in the stomach [11,12]. This knowledge gap emphasizes the necessity for further in-depth study in this field. Understanding the physiological processes involved in food intake regulation is critical for designing effective weight control and metabolic health strategies that consider the role of the stomach.

It is important to understand that food intake control involves a complex regulation by peripheral and central organs, highlighting gastrointestinal communication with the central nervous system (CNS). Some hormones of the gastrointestinal tract are secreted when responding to food intake, such as cholecystokinin (CCK) and glucagon-like peptide 1 (GLP-1). These hormones act on the hypothalamus, stimulating satiety, activating the anorexigenic pro-opiomelanocortin/cocaine–amphetamine-regulated transcript (POMC/CART) neurons, and inhibiting orexigenic agouti-related peptide/neuropeptide Y (AgRP/NPY) in the hypothalamus. Moreover, adipose tissue also participates in this regulation by secreting leptin, which acts on its receptor (ObRb) in the hypothalamus, stimulating the same pathway [13]. Thyrotropin-releasing hormone (TRH) and its receptors are also expressed in the hypothalamus and are involved in energy homeostasis in the hypothalamus–pituitary–thyroid pathway and are coexpressed with feeding regulatory peptides and receptors such as POMC/CART and NPY-Y [14].

This research attempts to close this knowledge gap by evaluating the functional consequences of acute excesses of various oils in the stomach on food intake management. By advancing our understanding of the physiological mechanisms involved in food intake control, we hope to provide valuable insights for the development of therapeutic approaches and more effective weight management strategies.

## 2. Materials and Methods

### 2.1. Animals and Supplementation Protocol

Rattus norvegicus of the Wistar lineage, male, young adult, from the vivarium of the Agricultural Sciences Center of the Federal University of Piauí were used. The animals were kept in a room with a controlled temperature of 22 °C, humidity, and a light–dark photoperiod (12:12 h), with free access to water and rodent food (Presence/Food Nutrition/Paulínea—SP). After each experiment, the animals were euthanized by the anesthetic deepening of the xylazine/ketamine cocktail (ketamine, 200 mg/kg, and xylazine, 20 mg/kg), administered via intraperitoneal.

Previously, the animals were monitored in the same way. The contents of the gavage, including the sources of oils and the saline solution, were unknown to the researcher who conducted the experiments. The study was blinded.

The FA sources used in our in vivo experiments were extra virgin olive oil (EVOO) (rich in oleic acid, C18:1), golden linseed oil (GLO) (rich in α-linolenic acid, ω3, C18:3), soybean oil (SO) (rich in linoleic acid, ω6, C18:2), and palm oil (PO) (rich in palmitic acid, C16:0). Samples of each oil were analyzed for FA composition by gas–liquid chromatography (GC-6850 Series Gas Chromatography System; Agilent Technologies). The samples of each oil source (50 µL) were saponified and esterified following Hartman and Lago’s (1973) procedures 1. At the end of the process, 2 mL of hexane was added and vortexed for 30 s. After, 5 mL of saturated NaCl was added, vortexed for 30 s, and left to rest until phase separation. The supernatant was recovered and dried using nitrogen. Samples were resuspended with 150 uL of hexane and followed by the gas chromatograph. The chromatographic running was carried out using a GCMS-QP2010 from Shimadzu (Tokyo, Japan), with a silica column Stabilwax (30 m × 0.25 mm, and 0.25 μm of internal diameter) purchased from Restek^®^. Ultrapure helium was adopted as running gas (1.3 mL/min). Using an automatic injector (AOC-20i), 1 mL of the samples was injected, in a ratio of 1:10 (split). The chromatographic conditions followed those of Cintra et al. (2006), 2, established as 250 °C of injector temperature, oven beginning at 80 °C following 5 °C/min until 175 °C, and 3 °C/min until 230 °C, maintaining for 20 min. The conditions of the mass spectrometer were established as ionization voltage 70 eV, ionization source at 200 °C, maintaining full scanning mode with amplitude between 35 and 500 *m*/*z*, and 0.2 s by scanning [15,16].

The fatty acid composition of the oils is shown in Table 1. In our study, the sources of fatty acids were manipulated through acute oral supplementation (gavage), at a dose of 0.8 mL/100 g. We chose this dose of 0.8 mL/100 g based on previous research by McConnell et al. (2008) [17], which determined the rat stomach volume. The authors recommended a maximum dosage of 10 mL/kg; thus, we adopted a safety margin of 20% lower, resulting in 0.8 mL/100 g. We also made a dose–response curve, with the doses 0.6 mL/100 g, 0.8 mL/100 g, and 1.0 mL/100 g on the expression of AgRP and POMC genes, with the better response being 0.8 mL/100 g. All treatments were performed according to the plan accepted by the Committee on Ethics in the Use of Animals (CEUA) of the Federal University of Piauí, No. 25/15.

### 2.2. Experimental Design

The animals were initially sorted in descending order of body weight, and then the groups were assigned based on a randomly generated sequence. The allocation of animals into the experimental groups followed the same cycle repeatedly until the desired number of animals was reached. After distributing the animals into each group in a randomized manner, the standard error and coefficient of variation (CV) were calculated, with the CV of each group not exceeding 10%.

The experiment was divided into two studies: Study 1, where the animals had an intact vagus nerve, and Study 2, where the animals underwent subdiaphragmatic vagotomy surgery.

In Study 1, the animals were randomly distributed into 5 groups and received acute supplementation by gavage with saline solution (CON), EVOO, GLO, SO, and PO, and the following parameters were evaluated: food consumption during 24 h, gastrointestinal transit from the collection of feces (24 h). After receiving treatment with different oils, one group received a test meal 30 min later. Another group received a test meal 180 min later to evaluate gastric emptying, as it represents the early and late times standardized in previous studies to verify the effect on gastric emptying [18,19]. Furthermore, after these same times, the hypothalamus was collected for analysis of the expression pattern of the hypothalamic genes: neuropeptide Y (NPY), pro-opiomelanocortin (POMC), cocaine- and amphetamine-regulated transcription (CART), glucagon-like peptide-1 receptor (GLP-1R), cholecystokinin receptor (CCKR), leptin receptor (ObRbR), and hormone thyrotropin releaser (TRH).

To assess the involvement of parasympathetic innervation in the communication between the GI tract and the hypothalamus, we developed Study 2 to mediate the effects of different sources of AG. The animals underwent subdiaphragmatic vagotomy surgery, according to De Sousa Cavalante et al. (2020) [20]; here, briefly, a 1 cm transverse right abdominal incision was performed 0.5 cm below the xiphisternal from the linea alba. The liver was carefully retracted with a small cotton pellet moistened with sterile normal saline, and the costal arch was pulled using a vascular clamp to expose the esophagus, where the dorsal and ventral branches of the vagus nerve were exposed along the subdiaphragmatic esophagus.

Then, after 4 days of recovery, independent protocols were initiated, such as removal of the hypothalamus (30 min after supplementation with the sources of AG, EVOO, and PO) to evaluate the pattern of expression of hypothalamic orexigenic and anorexigenic genes, as well as evaluation of feed consumption in 24 h and gastrointestinal transit (24 h) from the evaluation of fecal volume. The choice of EVOO and PO for Study 2 was based on the observation that EVOO had greater effects on gastric function, favoring a reduction in food consumption, while the effects of PO were antagonistic about these variables. In addition, PO exerted greater influences on the expression of anorexigenic and orexigenic neuropeptides. All tests were performed as described below.

### 2.3. Assessment of Dietary Intake and Stool Weight

Dietary intake was evaluated throughout the 4 days, 2 days before the oil administration, on the day of administration, and on the day after oil gavage. The rats were placed in individual cages, and the feed and water intake were monitored daily between 9:00 and 10:00 AM. Each rat had free daily access to filtered water and rodent food (Presence/Food Nutrition/Paulínea—SP), and the amount consumed over a 24 h interval was quantified. Also, the stool weight during this period was measured.

### 2.4. Gastric Emptying of Liquids

Gastric emptying of liquids was performed by a modification of the dye dilution technique described by Reynell and Spray (1956) [21]. The animals were fasted for 16 h, with free access to water. After the fasting period, the rats were randomly distributed into 5 groups and subjected to overload by gavage with saline solution and the four sources of AG. After 30 or 170 min of the different FA sources’ gavage, the animals received a new gavage of 1.5 mL of 5% phenol red glucose solution (0.75 mg/mL). Then, after an additional 10 min, the animals were euthanized through the intraperitoneal (IP) administration of an overdose of the xylazine/ketamine cocktail.

Gastric retention was analyzed according to Silva et al. (2021) [22]. After surgical exeresis, the gut was rapidly ligated at the esophagus, duodenum, and terminal ileum. It was then divided into consecutive segments: stomach, proximal small intestine (~40%), mid–small intestine (~30%), and distal small intestine (~30%). The volume of each segment was evaluated by placing it in a graduated cylinder that contained 100 mL of 0.1 N NaOH solution. The segments were then cut and homogenized with a mixer for 30 s. The suspension was allowed to settle for 20 min, and 10 mL of the supernatant was centrifuged for 10 min at 2800 rpm. Proteins were precipitated with 0.5 mL of 20% trichloroacetic acid solution. After centrifugation, 3 mL of the supernatant was added to 4 mL of 0.5 N NaOH, and the samples were read by a spectrophotometer at 560 nm to construct dilution curves by plotting the dye concentrations against the optical densities (ODs). The linear coefficient (α) of the dilution curve defined the solution concentration (C = OD) and the amount of phenol red (m) that was recovered from each segment (m = C × volume). Fractional gastric dye recovery values are expressed as follows: gastric dye recovery (%) = 1 − (amount of phenol red recovered in stomach/total amount of phenol red recovered from all segments) × 100.

### 2.5. Procedure for Collecting the Hypothalamus

Initially, the animals fasted for 16 h with free access to water. Following the gavages, the animals in each group were killed at various collecting times. Hypothalamus collection was taken by the method of Morris and Pavia (2004) [23], in which the brain was rapidly removed and the whole hypothalamus was dissected on ice by making two coronal cuts along the hypothalamic sulcis. The hypothalamus was removed by making vertical incisions along its lateral edge. Following that, the hypothalamus was promptly taken for investigation of the hypothalamic gene expression profile. The hypothalamus was stored in a microtube in a freezer at −80 °C after collection for subsequent examination using real-time polymerase chain reaction (PCR).

### 2.6. RNA Extraction from Hypothalamic Tissue

The hypothalamus was homogenized, and RNA was extracted using Trizol^®^ reagent (Ambion, Carlsbad, CA, USA) according to the manufacturer’s instructions. An amount of 2 µg of total RNA was used for each RT reaction, and cDNA synthesis was performed using the SuperScript™ First-Strand Synthesis System (Applied Biosystems, Forest City, CA, USA) and random primers p(dN)6 (Roche, Mannheim, Germany). Real-time polymerase chain reaction (qPCR) was performed using the QuantStudio 12 K Flex Real-Time PCR System (Applied Biosystems, Carlsbad, CA, USA) using Power SYBR Green PCR Master Mix (Applied Biosystems, Warrington, UK). Specific primers were designed for each target gene according to sequences taken from GenBank or the literature. NPY (sense: TGTGTTTGGGCATTCTGG; antisense: GCTGGATCTCTTGCCATATC), AgRP (sense: GCAGAGGTGCTAGATCCACAGAA; antisense: AGGACTCGTGCAGCCTTACAC), POMC (sense: ATAGACGTGTGGAGCTGGTGC; antisense: GCAAGCCAGCAGGTTGCT), CART (sense: CGCTGTGTTGCAGATTGAAGC; antisense: AGCGTCACACATGGGGACTT), CCKR (sense: AGAGCTAAGTGGGACTTCACTG; antisense: GCCATCTCCAATTCTCTGGT), GLP1R (sense: ACAGGTCTCTTCTGCAACCG; antisense: ATGCCCTTGGAGCACACTAC), ObRbR (sense: CAGTACCCAGAGCCAAAGT; antisense: GGCTTCACAACAAGCATGG), TRH (sense: CGACCCTGGATTCGGGAGTAT; antisense: CTGGAGTCTGCGAAGTGGAGA).

The internal control gene-stability measures were calculated using the GeNorm tool. It was determined that the most stable housekeeping gene for the present study was HPRT. Relative quantification of mRNA was calculated by 2^−ΔΔCT^. Data were normalized to β-2M expression and reported as fold changes compared to values obtained from the control group as previously described in other works from our group [24,25].

### 2.7. Procedure for Performing Subdiaphragmatic Vagotomy

Animals were fasted for 16 h, with free access to water. After anesthesia with the ketamine/xylazine cocktail (90 mg/kg and 10 mg/kg), the rats were submitted to median laparotomy and exposure of the abdominal esophagus followed by subdiaphragmatic truncal vagotomy, which was performed through esophageal vagotomy at 1.0–1.5 cm above the cardia and application of 70% alcohol according to the technique of da Graça et al. (2015) [26]. Next, the rats were kept in individual cages with free access to water and food. After the vagotomy procedure, the animals were kept in a recovery period for 72 h.

### 2.8. Outcome Measures

The two studies’ outcomes were as follows: food consumption, stool weight, gastric retention rate, and expression of genes related to food intake control. We also compared the effects of oil supplementation in non-vagotomized rats vs. vagotomized rats.

### 2.9. Data Analysis

Shapiro–Wilk test was employed to assess data normality, and the results of each group (n = 8–10) were expressed as the mean ± SD. Then, for the purpose of comparison between the groups, a one-way analysis of variance (ANOVA) was performed, followed by Dunnett’s post-test. Also, a two-way ANOVA was performed, followed by Sidak’s post-test. The difference was considered significant if the *p*-value was <0.05, adopting a 95% confidence interval. Statistical analysis was performed in GraphPad PRISM 9.

## 3. Results

Figure 1 shows the experimental design (panel A), relative food intake (panel B; g/100 g), and stool weight (panel C; g/100 g) over 4 days in the CON, EVOO, GLO, SO, and PO groups. Days −2 and −1 precede the therapy with various fatty acids (FAs). The third day saw the administration of saline or FAs. Day +1 is the day after the acute treatment. When compared to the control group, EVOO and GLO (panel B) had reduced (*p* < 0.05) food consumption. Furthermore, only EVOO (panel C) lowers the stool weight (*p* < 0.05) when compared to the CON.

Figure 2 shows the gastric retention after 10 min postprandial in different oil groups treated in 30 min (panel C) or 180 min (panel C) on the CON, EVOO, GLO, SO, and PO groups. Thirty minutes after EVOO and GLO treatment, we observed an increase in gastric retention (*p* < 0.05) compared to CON. There is no difference between groups after 180 min. After 30 min, the EVOO group had increased gastric retention (*p* < 0.05). After 30 min, there is a significant increase (*p* < 0.05) in GLO-induced gastric retention. After 180 min, the PO group experienced a significant increase in gastric retention (*p* < 0.05).

Figure 3 compares the hypothalamic mRNA expression of NPY (panel B), POMC (panel C), CART (panel D), TRH (panel E), GLP1R (panel F), ObRb (panel G), and CCK (panel H) after 30 min in the control (CON), extra virgin olive oil (EVOO), gold linseed oil (GLO), soy oil (SO), and palm oil (PO) groups. Treatment with PO increased (*p* < 0.05) the hypothalamic expression of neuropeptide Y (NPY). CART mRNA was significantly lower in the PO-treated rats compared to the CON animals (*p* < 0.05). The levels of GLP1R mRNA in the EVOO, GLO, and SO groups are all substantially lower than in the CON group (*p* < 0.05). After 30 min of intervention with the aforementioned FAs, we found no significant change in POMC, TRH, or CCK gene expression across the groups (*p* > 0.05).

The expression levels of NPY (panel B), POMC (panel C), CART (panel D), TRH (panel E), GLP1R (panel F), ObRb (panel G), and CCK (panel H) in the hypothalamus were analyzed after 180 min in the control (CON), extra virgin olive oil (EVOO), gold linseed oil (GLO), soy oil (SO), and palm oil (PO) groups, as shown in Figure 4. Upon administration of SO or PO to rats, a statistically significant increase (*p* < 0.05) in the expression of POMC was observed. In comparison to CON, the mRNA expression of CART was observed to be lower in SO and PO (*p* < 0.05). Following 180 min, no significant alterations were observed in the mRNA gene expression of TRH, GLP1R, ObRbR, or CCK among the intervention groups subjected to FAs (*p* > 0.05).

Figure 5 shows the relative food consumption (panel B; g/100 g) and stool weight (panel C; g/100 g) of vagotomized rats in the control (CON), extra virgin olive oil (EVOO), and palm oil (PO) groups for 5 days. Days −2 and −1 precede the treatment with different fatty acids (FAs). On the third day, saline or FAs were administered. Day +1 represents the day after the acute treatment. We observed that EVOO and PO have a reduced (*p* < 0.05) food intake when compared to CON, and the food intake in EVOO was also smaller (*p* < 0.05) when compared to the PO group. There is no difference in stool weight (*p* > 0.05). Panels D, E, and F depict the differences in food consumption between non-vagotomized and vagotomized animals for CON (panel D), EVOO (panel E), and PO (panel F). We found that the EVOO group consumed more food after vagotomy (*p* < 0.05), while the PO group consumed less (*p* < 0.05).

Figure 6 shows the relative expression of the hypothalamic mRNAs of NPY (panel B), POMC (panel C), CART (panel D), TRH (panel E), ObRb (panel F), and CCK (panel G) after 30 min in the CON, EVOO, and PO groups. We found that rats treated with EVOO and PO had lower (*p* < 0.05) NPY expression in the hypothalamus when compared to the controls.

## 4. Discussion

Our discoveries indicate that food intake changes depending on the type of FAs consumed. Firstly, it adds an unexpected facet to the physiological effects of various FA types. Different types of FAs, including monounsaturated fatty acids (MUFAs) in EVOO and polyunsaturated fatty acids (PUFAs) in GLO, can suppress food intake by promoting gastric retention and increasing satiety. Additionally, EVOO promotes a decrease in gastrointestinal transit. Secondly, our study extends the repertoire of actions of different sources of FAs on the involvement of the parasympathetic nervous system, as different categories of FAs can influence feeding patterns in various ways. Therefore, the same source of FAs has various effects on food intake depending on vagal system function.

In addition, we demonstrated that SO and PO alter anorexigenic and orexigenic gene patterns in the hypothalamus without significantly affecting overall food consumption. Additionally, these oils increase gastric retention. Notably, PO decreases food intake in vagotomized rodents, suggesting an influence on parasympathetic mechanisms.

The increased gastric retention and reduced food intake with PO in vagotomized rodents can be attributed to the interaction of gastric fullness and satiety signals with hypothalamic appetite regulation. Gastric distention during meals activates vagal afferents, influencing the hypothalamic expression of orexigenic AgRP and anorexigenic POMC genes. This mechanism aligns with findings in Atlantic salmon, where fasting upregulates hypothalamic agrp1, indicating AgRP1 as an orexigenic signal [27]. Obesity-related changes in vagal afferent function promote food intake, highlighting the vagal pathway’s role in energy balance [28].

Bariatric surgery alters the methylation levels of specific CpG sites in AGRP, GHRL, and POMC genes, correlating with fat mass loss [29]. These changes significantly impact energy homeostasis and appetite control [30]. The observed decrease in food intake with PO in vagotomized rodents likely results from modifications in the parasympathetic regulation of these genes, influencing feeding behavior and gastric retention.

Other factors, such as LEAP2, also regulate food intake. LEAP2 acts as an antagonist to the growth hormone secretagogue receptor (GHSR) and suppresses ghrelin metabolic actions [31]. Neuronal populations in the hypothalamic arcuate nucleus (ARC), including AgRP/NPY and POMC, are crucial for energy homeostasis and appetite regulation [32]. NPY from AGRP neurons drives feeding behavior and energy homeostasis. In fish, the receptor NPY8R regulates feeding [33]. These findings suggest multiple factors, including LEAP2, AgRP/NPY, POMC, and NPY8R, are involved in the complex regulation of food intake [34].

Our results show that SO and PO affect gastric retention and hypothalamic gene expression without significantly changing overall food consumption. The paradoxical reduction in food intake by PO in vagotomized rodents underscores the complexity of parasympathetic mechanisms in appetite regulation. These findings provide new insights into the roles of dietary oils in energy homeostasis and suggest potential targets for obesity treatment through the modulation of hypothalamic pathways and vagal function.

Prior research investigated whether dietary nutrients modulate satiety and appetite signals and how various forms of fat affect gastric emptying and hypothalamic signaling [35,36]. Our study is a pioneer in the evaluation of different oils’ overload, rich in certain classes of fatty acids that are commonly ingested by the population in rats’ food behavior. The unique composition of each oil leads to different responses in gastrointestinal fat-sensing pathways, endocrine regulation, and gut–brain communication [37,38].

After 24 h, acute excess with EVOO and GLO decreased food intake in our study. EVOO contains over 80% MUFAs, specifically oleic acid, also known as ω9, whereas GLO contains over 60% PUFAs, of which 52% are α-linolenic acid, or ω3, and 20% are ω9 [39]. Sun et al. (2019) [40] found that the presence of unsaturation can alter gastrointestinal hormone release and influence food intake.

This phenomenon is more pronounced when comparing the EVOO and PO groups, as the EVOO group experienced a significant decrease in food intake in 24 h. More than or equal to 55% of PO consists of saturated fatty acids (SFAs), particularly palmitic oil and stearic oil, and 50% ω9 [41]. Concerning SO, the FA composition consists of 52% linoleic acid, or ω, another PUFA, 25% ω9 (MUFA), and additional PUFAs and SFAs [41]. Previous research indicates that PUFAs are more satiating than MUFAs and SFAs [42].

According to these, after 30 min, we observed that EVOO and GLO increased gastric retention compared to CON. EVOO and GLO are rich in PUFAs and MUFAs with 18 carbons. After 180 min, all groups experienced comparable gastric retention. In addition, the SO and PO exhibited a more pronounced effect after 180 min as opposed to 30 min.

Long-chain fatty acids (LCFAs) detected by the gut sensors can also affect the rate of gastric emptying, as can the length of the carbon chains of FAs. GPR50 and GPR120 stimulate the signaling of G proteins and the secretion of gastrointestinal hormones, including CCK, GLP-1, and PYY. CD36 is another transporter that increases intracellular Ca2+ release to stimulate hormone secretion. These hormones activate enteric nervous system (ENS) responses, thereby delaying gastric emptying by operating on gastric motility [43].

We also evaluated the relative hypothalamic expression of genes implicated in the regulation of appetite control. After 30 min, PO increased the expression of NPY mRNA in this manner. The increase in NPY mRNA promotes food consumption because these neurons are associated with orexigenic activity in the hypothalamus [44]. A plausible explanation for this is that the effect of PO is mediated by vagal afferent fibers, as rats treated with PO exhibited decreased food intake and NPY mRNA relative expression following vagotomy. Palmitic acid, one of the most abundant components of PO, was associated with inflammation via the activation of toll-like 2 and 3 receptors (TLR2 and TLR3) and nuclear factor kappa b (NF-κB)-mediated cytokine synthesis, insulin resistance, and NPY expression [45].

After vagotomy, the EVOO group presented a lower food intake compared to CON. The effects seem to also be mediated by a drastic reduction in NPY expression after vagotomy in rats with an acute overload of EVOO when compared with non-vagotomized rats. This suggests that EVOO effects in NPY are not mediated by vagal signaling, and a possible explanation is that oleic acid can cross the blood–brain barrier with direct effects on central food intake control. Furthermore, several FAs can cross the blood–brain barrier, due to the presence of fatty acid transport proteins 1 and 4 (FATP1, FATP4) and CD36 [46].

Similar to our results for consumption with EVOO, Oh et al. (2016) [47] found that intravenous infusion of oleic acid reduces food intake, and observed that vagotomy did not change the effect of oleic acid on reducing food intake; however, this study did not evaluate the effects of infusion on gastric function.

In addition, it is important to highlight that the food intake observed for EVOO after vagotomy was higher than that observed in non-vagotomized patients, possibly due to the elimination of the effects of gastric stretching, after delaying gastric emptying, via vagal afferents, thus contributing to higher food intake, further highlighting the local effects attributed to EVOO on the gastrointestinal tract.

This study has some limitations, such as the absence of a chronic evaluation of FAs intake and determination of gastrointestinal hormone concentrations. The employment of an overnight period of abstaining from food intake is another limitation, but it is justified as a method to alleviate the impact of differential food consumption on the expression of genes in the hypothalamus. However, we acknowledge that fasting itself can introduce complexities in the interpretation of findings. It is crucial to emphasize that fasting during the night is commonly employed in physiological experiments to challenge the functioning of the body through specific stimuli. A notable instance of this approach is observed in the administration of oral glucose tolerance tests and insulin tolerance tests.

When evaluating the entirety of the hypothalamus, we are faced with a situation that presents advantages and challenges. It is of utmost importance, however, to highlight that the primary benefit lies in its capacity to offer a comprehensive comprehension of molecular alterations in the hypothalamus [48,49,50,51,52]. This allows us to comprehend the overall panorama of interactions among different hypothalamic nuclei and their contribution to the functioning of the hypothalamus as a whole. This methodology seamlessly aligns with the fundamental nature of all experiments carried out in our investigation, which are firmly rooted in the exploration of global physiological phenomena. Nevertheless, we acknowledge that the examination of individual hypothalamic nuclei uncovers an extensive array of details, enabling the identification of distinct cellular classifications and their specific functions. This perspective enhances our comprehension by unraveling the intricate mechanisms that underlie fundamental physiological processes. Therefore, while both approaches possess their respective merits, we have opted for a more comprehensive standpoint, in accordance with the scope of this study, while simultaneously recognizing the valuable contribution that the analysis of individual hypothalamic nuclei can offer for more focused investigations in the future.

But some strengths also can be highlighted: we standardized the volume of oil administered to rats, to avoid the effects of gastric distension altering the velocity of gastric emptying.

In our research, we discovered that different types of oils have distinct impacts on variables related to food consumption. EVOO, in particular, reduced food intake, primarily through mechanisms that affect the gastrointestinal tract. Consequently, EVOO is suggested as a therapeutic adjunct in weight loss treatments. However, additional research is required to assess the long-term effects of this oil from the same perspective.

Although PO reduces food consumption in the absence of an autonomic response, it would not be recommended as it contributes to the development of peripheral and central inflammatory processes, systemic resistance to hormones such as insulin and leptin, and apoptosis of hypothalamic neurons, thus contributing to the development of obesity and cardiometabolic diseases.

## 5. Conclusions

This study discovered that different oils induce various effects on parameters related to food consumption. Specifically, EVOO reduces food consumption primarily through its effects on the gastrointestinal tract, making it a recommended adjunct for weight loss. Conversely, the intake of PO limits food consumption in the absence of an autonomic reaction, but it is not advised due to its contribution to the development of cardiometabolic disorders.

## Figures and Tables

**Figure 1 nutrients-16-02397-f001:**
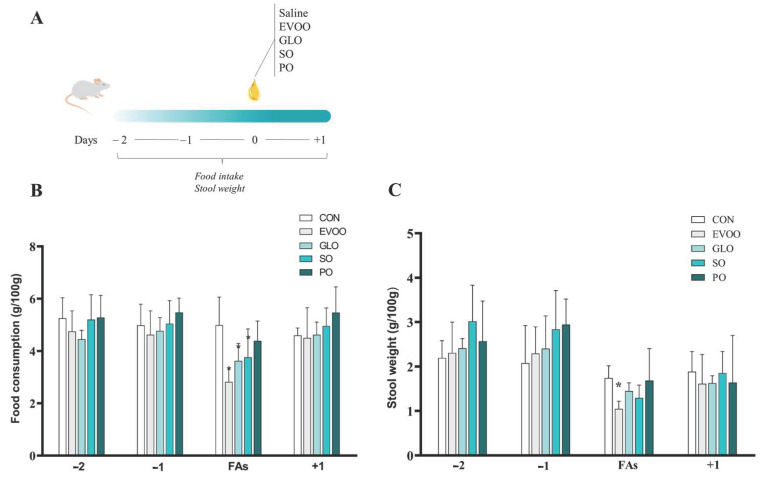
The influence of acute consumption of different oils in food intake (panel (**B**)) and stool weight (panel (**C**)) in 4 days. Legend: control group (CON); extra virgin olive oil (EVOO); gold linseed oil (GLO); soy oil (SO); palm oil (PO); fatty acids (FAs); Figure 1, panel (**A**) presents the experimental design; panel (**B**,**C**): −2 and −1: days before treatment with fatty acids; FAs: day of treatment with fatty acids; +1: day after the treatment with fatty acids. Data are expressed as mean ± SD and were statistically analyzed by two-way ANOVA, followed by the Sidak’s test. Significance: *p* < 0.05. * *p* < 0.05—vs. CON.

**Figure 2 nutrients-16-02397-f002:**
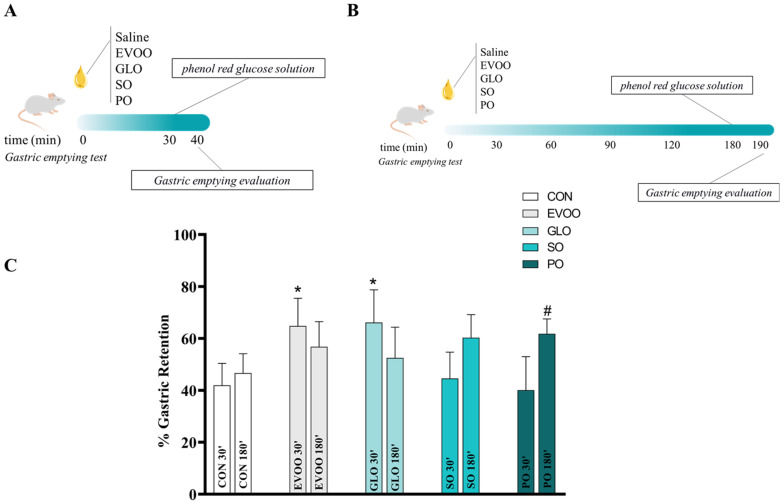
Gastric retention after 30 min (panel (**A**,**C**)) and 180 min (panel (**B**,**C**)) of intervention with fatty acids and differences between gastric retention evaluation times for extra virgin olive oil (EVOO), gold linseed oil (GLO), soy oil (SO), and palm oil (PO) groups. Legend: control group (CON); extra virgin olive oil (EVOO); gold linseed oil (GLO); soy oil (SO); palm oil (PO). Panels (**A**,**C**) present the experimental design. Data are expressed as mean ± SD and were statistically analyzed by two-way ANOVA, followed by the Sidak’s test. Significance: *p* < 0.05. * *p* < 0.05—vs. CON 30′; # *p* < 0.05—vs. CON 180′.

**Figure 3 nutrients-16-02397-f003:**
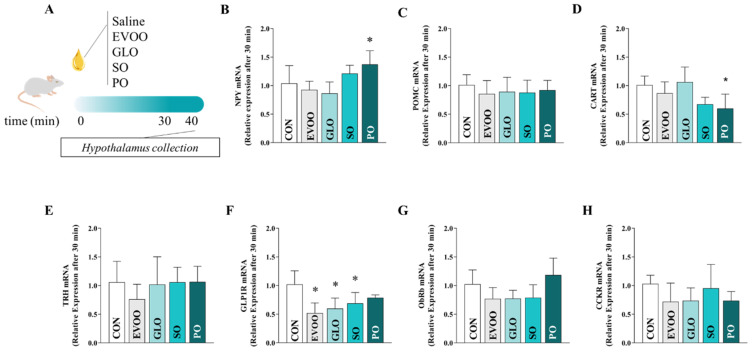
Relative expression of hypothalamic mRNA of NPY (panel (**B**)), POMC (panel (**C**)), CART (panel (**D**)), TRH (panel (**E**)), GLP1R (panel (**F**)), ObRb (panel (**G**)), and CCK (panel (**H**)) in the control group (CON), extra virgin olive oil (EVOO), gold linseed oil (GLO), soy oil (SO), and palm oil (PO) groups after 30 min. Legend: control group (CON); extra virgin olive oil (EVOO); gold linseed oil (GLO); soy oil (SO); palm oil (PO); neuropeptide Y (NPY); pro-opiomelanocortin (POMC); cocaine- and amphetamine-regulated transcript (CART); thyrotropin-releasing hormone (TRH); glucagon-like peptide-1 receptor (GLP1R); leptin receptor (ObRb); cholecystokinin (CCK). Panel (**A**) presents the experimental design. Data are expressed as mean ± SD and were statistically analyzed by one-way ANOVA, followed by the Dunnett’s test. Significance: *p* < 0.05. * *p* < 0.05—vs. CON.

**Figure 4 nutrients-16-02397-f004:**
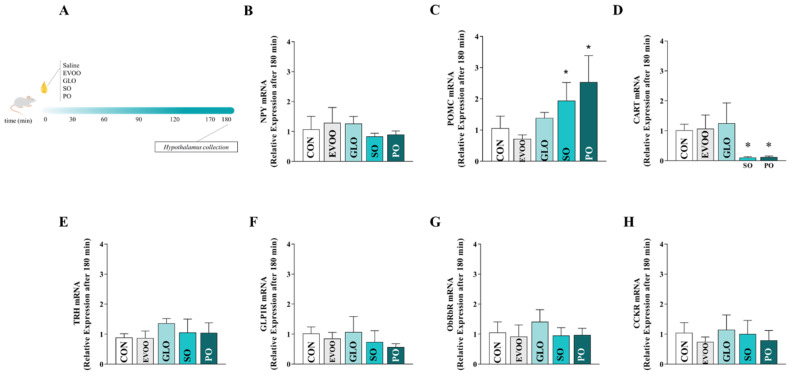
Relative expression of hypothalamic mRNA of NPY (panel (**B**)), POMC (panel (**C**)), CART (panel (**D**)), TRH (panel (**E**)), GLP1R (panel (**F**)), ObRb (panel (**G**)), and CCK (panel (**H**)) in the control group (CON), extra virgin olive oil (EVOO), gold linseed oil (GLO), soy oil (SO), and palm oil (PO) groups after 180 min. Legend: control group (CON); extra virgin olive oil (EVOO); gold linseed oil (GLO); soy oil (SO); palm oil (PO); neuropeptide Y (NPY); pro-opiomelanocortin (POMC); cocaine- and amphetamine-regulated transcript (CART); thyrotropin-releasing hormone (TRH); glucagon-like peptide-1 receptor (GLP1R); leptin receptor (ObRb); cholecystokinin (CCK). Panel (**A**) presents the experimental design. Data are expressed as mean ± SD and were statistically analyzed by one-way ANOVA, followed by the Dunnett’s test. Significance: *p* < 0.05. * *p* < 0.05—vs. CON.

**Figure 5 nutrients-16-02397-f005:**
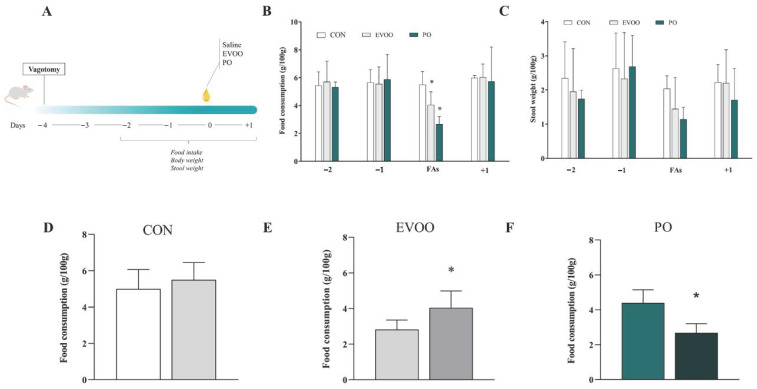
Relative food intake (Panel (**B**); g/100 g) and the stool weight (Panel (**C**); g/100 g) in the control group (CON), extra virgin olive oil (EVOO), and palm oil (PO) groups in 24 h and difference in food consumption of non-vagotomized animals and vagotomized animals for CON (Panel (**D**)), EVOO (Panel (**E**)), and PO (Panel (**F**)). Legend: control group (CON); extra virgin olive oil (EVOO); palm oil (PO); data are expressed as mean ± SD, and were statistically analyzed by one-way ANOVA, followed by the Dunnett’s test (Panel (**B**,**C**)) or by Student’s *t*-test (Panel (**D**–**F**)). Significance: *p* < 0.05. Panel (**A**) presents the experimental design. Panel (**B**,**C**): * *p* < 0.05—vs. CON. Panel (**D**–**F**): * *p* < 0.05—vs. non-vagotomized rats.

**Figure 6 nutrients-16-02397-f006:**
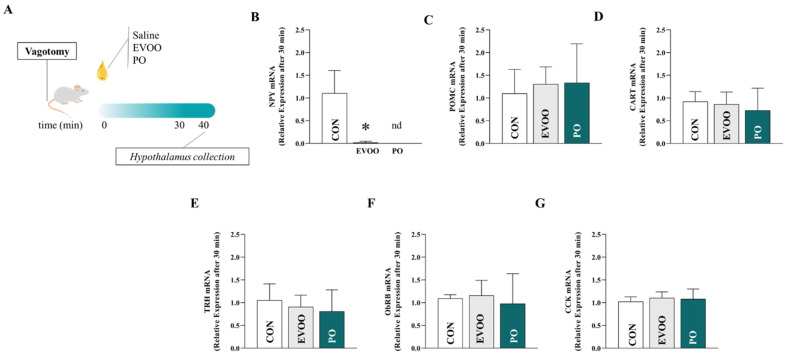
Relative expression of hypothalamic mRNA of NPY (panel (**B**)), POMC (panel (**C**)), CART (panel (**D**)), TRH (panel (**E**)), ObRb (panel (**F**)), and CCK (panel (**G**)) in the control group (CON), extra virgin olive oil (EVOO), and palm oil (PO) groups after 30 min. Legend: control group (CON); extra virgin olive oil (EVOO); palm oil (PO); neuropeptide Y (NPY); pro-opiomelanocortin (POMC); cocaine- and amphetamine-regulated transcript (CART); thyrotropin-releasing hormone (TRH); leptin receptor (ObRb); cholecystokinin (CCK). Panel (**A**) presents the experimental design. Data are expressed as mean ± SD and were statistically analyzed by one-way ANOVA, followed by the Dunnett’s test. Significance: *p* < 0.05 * *p* < 0.05—vs. CON.

**Table 1 nutrients-16-02397-t001:** Fatty acid profile of the evaluated oils (g/100 g of oil).

FAs	Nomenclature	EVOO	GLO	SO	PO
C12:0	Lauric	0.04	0.04	0.03	0.07
C14:0	Myristic	0.07	0.09	0.09	0.08
C15:0	Pentadecanoic	0.08	0.04	0.03	-
C16:0	Palmitic	5.65	6.20	10.85	40.47
C16:1	Palmitoleic	0.81	0.10	0.11	-
C17:0	Margaric	0.10	0.07	0.08	-
C17:1	Heptadecanoic	0.08	0.06	0.06	-
C18:0	Stearic	2.19	4.93	3.39	5.02
C18:1 Trans	Elaidic	-	-	-	0.07
C18:1 (w9)	Oleic	82.51	19.80	24.82	40.38
C18:2 Trans	Linoelaidic	0.09	0.08	0.28	0.12
C18:2 (w6)	Linoleic	6.39	15.28	52.72	10.16
C18:3 Trans	-	0.00	0.23	0.75	-
C18:3 (w3)	Linolenic	0.79	52.5	5.69	1.26
C20:0	Araquidic	0.53	0.16	0.32	1.06
C20:1 (w9)	Eicosenoic	0.34	0.12	0.21	0.34
C22:0	Docosanoic	0.19	0.16	0.40	0.68
C24:0	Tetracosanoic	0.14	0.14	0.17	-
Total SAT		8.99	11.83	15.36	47.38
Total MONO		83.74	20.08	25.2	40.79
Total POLI		7.27	68.09	59.44	11.54

## Data Availability

The original contributions presented in the study are included in the article, further inquiries can be directed to the corresponding authors.

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
