# Peer review of "Brain Perception of Different Oils on Appetite Regulation: An Anorectic Gene Expression Pattern in the Hypothalamus Dependent on the Vagus Nerve"

_nutrients, 2024, doi:10.3390/nu16152397_

Round 1
Reviewer 1 Report
Comments and Suggestions for Authors
The manuscript requires a number of chnages prior to publication:
1) Abstract - please, change
a) the description of methods into The effect of acute administration of olive oil (EVOO), linseed oil (GLO), soybean oil (SO), and palm oil (PO) on ......... was studied in male Wistar rats.
b) description of results - add precise info about gene expression alterations
3) Methods - please, explain the differences in observation time - e.g. after vagotomy - the experiment was carried out for 5 dyas, whereas control was observed for 3 or 4 days???
2) Results section
a) Figure 1 does not fit into the page; the title is not informative; should sound e.g. as The influence of .... on......
b) The description should directly under the figure
c) in description - instead of Figures 1A etc. it should be written "panel A" etc.
d) similar comments concern other figures
3) Discussion section
Paragraphs 3 and 4 present a brief review of factors regulating the food intake;
however, the authors do not really discuss their findings in the light of other available data in the following 4 paragraphs; the issue comes back in paragraph 9.
Please, a) reorganize the discussion, b) inlcude the discussion of observed alterations i.e. decreased expression of diffrent molecules after different treatment and different time in correlation to findings from chronic experiments
Comments on the Quality of English Language
English requires intense editing. Below some examples:
line 131 - Another group received a test meal 180 minutes later, to evaluate gastric emptying, because represents the early and late times
140 - we developed Study 2, mediating the effects of different sources of AG
152 - The choice of EVOO and PO for Study 2 was because EVOO had greater effects
379 - in controlling feed intake
Author Response
The manuscript requires a number of changes prior to publication:
1) Abstract - please, change
- a) the description of methods into The effect of acute administration of olive oil (EVOO), linseed oil (GLO), soybean oil (SO), and palm oil (PO) on ......... was studied in male Wistar rats.
R: We agree with this comment and change the abstract. Line 32-34.
Before: Male Wistar Rattus norvegicus were utilized as the experimental subjects. We explored the outcomes of olive oil (EVOO), linseed oil (GLO), soybean oil (SO), and palm oil (PO);
After: The effect of acute administration of olive oil (EVOO), linseed oil (GLO), soybean oil (SO), and palm oil (PO) was studied in Male Wistar Rattus norvegicus.
- b) description of results - add precise info about gene expression alterations
R: We added more information about gene expression. Lines 37; 41.
Before: (3) Results: Both EVOO and GLO were found to enhance the rate of stomach retention, leading to a decrease in hunger. On the other hand, the reduction in food intake caused by SO was accompanied by delayed effects on stomach retention. Although PO increased stomach retention after 180 minutes, it did not affect food intake. We observed distinct effects on gene expression, with PO showing the most pronounced impact. It was subsequently verified that the absence of autonomic reaction did not nullify the influence of EVOO in reducing food consumption. Moreover, in the absence of parasympathetic responses, animals that received PO exhibited a significant decrease in food consumption;
After: (3) Results: Both EVOO and GLO were found to enhance the rate of stomach retention, leading to a decrease in hunger. On the other hand, the reduction in food intake caused by SO was accompanied by delayed effects on stomach retention. PO caused alteration in mRNA expression of NPY, POMC and CART. Although PO increased stomach retention after 180min, it did not affect food intake. It was subsequently verified that the absence of an autonomic reaction did not nullify the influence of EVOO in reducing food consumption. Moreover, in the absence of parasympathetic responses, animals that received PO exhibited a significant decrease in food consumption, probably mediated by lower NPY expression.
3) Methods - please, explain the differences in observation time - e.g. after vagotomy - the experiment was carried out for 5 days, whereas control was observed for 3 or 4 days???
R: The difference between the protocols in the two studies is due to the vagotomy surgery that needed two days for recovery. To minimize the effect of vagotomy on food intake and stool weight we waited two days to collect stools and mensurate food consumption (study 2).
2) Results section
- a) Figure 1 does not fit into the page; the title is not informative; should sound e.g. as The influence of .... on......
R: Done. The new title “Figure 01. The influence of acute consumption of different oils in food intake (panel B) and stool weight (panel C) in 4 days.”
- b) The description should directly under the figure.
R: Done.
- c) in description - instead of Figures 1A etc. it should be written "panel A" etc.
R: Done.
- d) similar comments concern other figures
R: Done.
3) Discussion section
Paragraphs 3 and 4 present a brief review of factors regulating the food intake;
however, the authors do not really discuss their findings in the light of other available data in the following 4 paragraphs; the issue comes back in paragraph 9.
Please, a) reorganize the discussion, b) include the discussion of observed alterations i.e. decreased expression of different molecules after different treatment and different time in correlation to findings from chronic experiments
R: We reorganize the aforementioned paragraph, modifying the discussion. Lines 326-446.
Comments on the Quality of English Language
English requires intense editing. Below some examples:
line 131 - Another group received a test meal 180 minutes later, to evaluate gastric emptying, because represents the early and late times
R: We agree. Thanks for you observation.
Before: Another group received a test meal 180 minutes later, to evaluate gastric emptying, be-cause represents the early and late times standardized in previous works to verify an effect in gastric emptying.
After: Another group received a test meal 180 minutes later to evaluate gastric emptying, as it represents the early and late times standardized in previous studies to verify the effect on gastric emptying.
140 - we developed Study 2, mediating the effects of different sources of AG
R: We agree. Thanks for you observation.
Before: To assess the involvement of parasympathetic innervation in the communication between the GI tract and the hypothalamus, we developed Study 2, mediating the effects of different sources of AG.
After: To assess the involvement of parasympathetic innervation in the communication between the GI tract and the hypothalamus, we developed Study 2 to mediate the effects of different sources of AG.
152 - The choice of EVOO and PO for Study 2 was because EVOO had greater effects
R: We agree. Thanks for you observation.
Before: The choice of EVOO and PO for Study 2 was because EVOO had greater effects on gastric function, favoring a reduction in food consumption, and the effects of PO were antagonis-tic in relation to these variables.
After: The choice of EVOO and PO for Study 2 was based on the observation that EVOO had greater effects on gastric function, favoring a reduction in food consumption, while the effects of PO were antagonistic in relation to these variables.
379 - in controlling feed intake
R: Done.
After: The orexigenic AgRP and anorexigenic POMC play crucial roles in controlling food in-take in vertebrates.
Reviewer 2 Report
Comments and Suggestions for Authors
The authors report the results of a murine study which assessed the cerebral perception of four plant-derived oils on the regulation of appetite. The oils had different impacts on factors relating to appetite. The study was carefully designed and suitably powered.
1. Were the scientists who dissected the animals and carried out the laboratory assessments blind to group status?
2. The authors did not include a control group (unless they wish to argue that one of the four oils acted as a control). They should include a paragraph justifying this.
3. Lines 103-4. More details are required of these analyses. Were they, for example, based on FAME analyses?
4. Line 143. The spelling "xiphisternal" is more common.
5. Line 393. I would argue that liver-expressed antimicrobial peptide 2 (LEAP2) has not been recently discovered. It was formally named LEAP-2 over 20 years ago.
Author Response
The authors report the results of a murine study which assessed the cerebral perception of four plant-derived oils on the regulation of appetite. The oils had different impacts on factors relating to appetite. The study was carefully designed and suitably powered.
- Were the scientists who dissected the animals and carried out the laboratory assessments blind to group status?
R: Yes, the experiment involved the acute administration of different oils. Previously, the animals were monitored in the same way. The contents of the gavage, including the sources of oils and the saline solution, were unknown to the researcher who conducted the experiments. The study was blinded. We added this information in lines 98-101.
- The authors did not include a control group (unless they wish to argue that one of the four oils acted as a control). They should include a paragraph justifying this.
R: We included a control group that was administrated saline solution. This information is in lines 126-127 and in all the figures.
- Lines 103-4. More details are required of these analyses. Were they, for example, based on FAME analyses?
R: We include more detail about the analyses. Thank you for your observations.
The samples of each oil source (50 µL) were saponified and esterified following Hartman & Lago (1973) procedures 1. In the end of the process, was added 2 mL of hexane and vor-texed for 30 sec. After, 5 mL of saturated NaCl, vortexed 30 sec and resting until phase separation. The supernatant was recovered and dried using nitrogen. Samples were re-suspended with 150 uL of hexane and followed to the gas-chromatograph. The chroma-tographic running was carried out using a GCMS-QP2010 from Shimadzu (Tokyo, Japan), with a silica column Stabilwax (30 m x 0.25 mm, and 0.25 m of internal diameter) pur-chased from Restek®. Ultrapure Helium was adopted as running gas (1.3 mL/min). Using an automatic injector (AOC-20i), 1 mL of samples were injected, in a ratio of 1:10 (split). The chromatographic conditions were following Cintra et al., (2006)2, established as 250 °C of injector temperature; oven beginning at 80 °C following 5 °C/min until 175 °C, and 3 °C/min until 230 °C, maintaining for 20 minutes. The conditions of mass spectrometer were stablished as ionization voltage 70 eV, ionization source was at 200 °C, maintaining full scanning mode with amplitude between 35-500 m/z, and 0.2 seconds by scanning.
- Line 143. The spelling "xiphisternal" is more common.
R: We agree. Thanks for you observation.
- Line 393. I would argue that liver-expressed antimicrobial peptide 2 (LEAP2) has not been recently discovered. It was formally named LEAP-2 over 20 years ago.
R: We agree. Thanks for you observation. We removed this information.
After: LEAP2 acts as an antagonist of the growth hormone secretagogue receptor (GHSR) and can suppress the metabolic actions of ghrelin [29].
Round 2
Reviewer 1 Report
Comments and Suggestions for Authors
Dear authors, thank you for the corrections. Minor comments:
1. In abstract - please, remove the abbreviation of gastric retention (GR), as it is not being used later on in abstract
2. In abstract - please, change the sentence "PO caused alteration in mRNA expression of NPY, POMC, and CART" into more detailed description
- develop the abbrevationes
- indicate the type of change - increse/decrese
- indicate the time after administration 30 or 180 min
Author Response
Dear authors, thank you for the corrections. Minor comments:
- In abstract - please, remove the abbreviation of gastric retention (GR), as it is not being used later on in abstract
R: We very much appreciate your contribution to this work. Done.
- In abstract - please, change the sentence "PO caused alteration in mRNA expression of NPY, POMC, and CART" into more detailed description
- develop the abbrevationes
- indicate the type of change - increse/decrese
- indicate the time after administration 30 or 180 min
R: Thanks for your observations. We changed this sentence to:
PO caused an increase mRNA expression of neuropeptide Y (NPY) and a decrease in cocaine- and amphetamine-regulated transcript (CART) after 30 minutes, but an increased pro-opiomelanocortin (POMC) mRNA expression after 180 minutes.
Reviewer 2 Report
Comments and Suggestions for Authors
I thank the authors for kindly amending their submission. This excellent paper is now suitable for publication.
Comments on the Quality of English LanguageSlight editing is required in the English - e.g. line 108.
Author Response
Comments and Suggestions for Authors
I thank the authors for kindly amending their submission. This excellent paper is now suitable for publication.
Comments on the Quality of English Language
Slight editing is required in the English - e.g. line 108.
R: We very much appreciate your contribution to this work. Thanks for your observation. We changed the phrase to this:
The samples of each oil source (50 µL) were saponified and esterified following the pro-cedures of Hartman & Lago [15]. At the end of the process, 2 mL of hexane was added and vortexed for 30 seconds.